# Fusaricidins, Polymyxins and Volatiles Produced by *Paenibacillus polymyxa* Strains DSM 32871 and M1

**DOI:** 10.3390/pathogens10111485

**Published:** 2021-11-15

**Authors:** Pascal Mülner, Elisa Schwarz, Kristin Dietel, Stefanie Herfort, Jennifer Jähne, Peter Lasch, Tomislav Cernava, Gabriele Berg, Joachim Vater

**Affiliations:** 1ABITEP GmbH, Glienicker Weg 185, 12489 Berlin, Germany; pascal.muelner@tugraz.at (P.M.); Schwarz@abitep.de (E.S.); Dietel@abitep.de (K.D.); 2Institute of Environmental Biotechnology, Graz University of Technology, Petergasse 12, 8010 Graz, Austria; tomislav.cernava@tugraz.at (T.C.); gabriele.berg@tugraz.at (G.B.); 3Robert Koch-Institut, ZBS6, Proteomics and Spectroscopy, Seestr 10, 13353 Berlin, Germany; HerfortS@rki.de (S.H.); JaehneJ@rki.de (J.J.); LaschP@rki.de (P.L.)

**Keywords:** *P. polymyxa*, bioactive peptides, volatiles, MALDI-TOF mass spectrometry, GC–MS

## Abstract

*Paenibacilli* are efficient producers of potent agents against bacterial and fungal pathogens, which are of great interest both for therapeutic applications in medicine as well as in agrobiotechnology. Lipopeptides produced by such organisms play a major role in their potential to inactivate pathogens. In this work we investigated two lipopeptide complexes, the fusaricidins and the polymyxins, produced by *Paenibacillus polymyxa* strains DSM 32871 and M1 by MALDI-TOF mass spectrometry. The fusaricidins show potent antifungal activities and are distinguished by an unusual variability. For strain DSM 32871 we identified numerous yet unknown variants mass spectrometrically. DSM 32871 produces polymyxins of type E (colistins), while M1 forms polymyxins P. For both strains, novel but not yet completely characterized polymyxin species were detected, which possibly are glycosylated. These compounds may be of interest therapeutically, because polymyxins have gained increasing attention as last-resort antibiotics against multiresistant pathogenic Gram-negative bacteria. In addition, the volatilomes of DSM 32781 and M1 were investigated with a GC–MS approach using different cultivation media. Production of volatile organic compounds (VOCs) was strain and medium dependent. In particular, strain M1 manifested as an efficient VOC-producer that exhibited formation of 25 volatiles in total. A characteristic feature of *Paenibacilli* is the formation of volatile pyrazine derivatives.

## 1. Introduction

*Paenibacillus* spp. are distinguished by a large arsenal of bioactive secondary metabolites, such as nonribosomally formed lipopeptides, lassopeptides, polyketides, lantibiotics and bacteriocines, which are of high importance both in agrobiotechnology and medicine [1,2,3,4]. They are efficient producers of potent agents against both bacterial and fungal pathogens. The most abundant bioactive compounds of these organisms are lipopeptides, which cover a broad range of structurally diverse linear and cyclic species with peptide chain lengths between 6 and 13 amino acids and manifold variations in their fatty acid chains. They comprise fusaricidins and the related LiF-antibiotics (cyclic lipohexapeptides connected with a guanidinilated ß-hydroxy fatty acid with chain lengths between 13–19 carbon atoms) [5,6,7,8,9,10,11,12], paenilipoheptins (cyclic lipoheptapeptides, ß-amino fatty acid) [13], octapeptins (cyclic lipooctapeptides, ß-hydroxy fatty acid) [1,14], polypeptins (cyclic lipononapeptides, ß-hydroxy fatty acid) [1,15], pelgipeptins (cyclic lipononapeptides, ß-hydroxy fatty acid) [1,16], polymyxins (cyclic lipodecapeptides, C8,9-unsubstituted fatty acids) [1,17], tridecapeptins (linear lipotridecapeptides, ß-hydroxy fatty acids) [1,18,19,20] and paenibacterins (cyclic lipotridecapeptides, pentadecanoic acid) [21,22]. A promising prospect for medical application of these agents is that some of them display strong activities against multidrug-resistant pathogens covering both Gram-negative as well as Gram-positive bacteria. Therefore, they may be a valuable resource for the development of novel antibiotics to overcome the increasing problem of antibiotic resistance. In particular, the exploitation of the bioactive agents produced by *P. polymyxa* may contribute to solve this insistent public health issue.

The present study is focused on *Paenibacillus polymyxa* strains, which are efficient plant growth-promoting rhizobacteria (PGPR) colonizing surfaces of plant roots in the rhizosphere [23,24,25]. They also appear as endophytes within the plant [26,27]. Such strains produce four classes of lipopeptides, the fusaricidins, paenilipoheptins, polymyxins and tridecaptins. All of them comprise numerous structural homologs. In particular, fusaricidins [10,11,12] show an unusual complexity of isoforms providing a large spectrum of closely related agents with graduated activities against fungal pathogens. In this work we investigated the fusaricidin and polymyxin complexes produced by *P. polymyxa* DSM 32871 in comparison to those formed by the well-characterized strain M1 as model organism. M1 is an efficient plant growth promoting rhizobacterium. Its complete genome was sequenced by Niu et al. [28] revealing nine gene clusters involved in nonribosomal synthesis of bioactive peptides and polyketides. DSM 32871 is a novel strain. Its genome sequence is not yet available.

Fusaricidins were discovered by Kakinuma et al. [5,6] and thereafter investigated by other authors in detail [7,8,9]. They provide a large reservoir of potent antifungal biocontrol agents against a broad array of phytopathogenic fungi, such as *Fusarium* strains [26,29], *Rhizoctonia solani* [25], *Leptosphaeria maculans* [30], *Sclerotinia sclerotiorum* [25] and *Botrytis cinerea* [25], for examples. They are commonly used to cure plant diseases, such as fusarium wilt of cucumber [29]; gibberella ear rot of maize [26]; blackleg disease of canola [30] or bacterial leaf blight in rice [31].

Polymyxins are cationic cyclic lipodecapeptides, which also appear as a family of closely related variants [17,32]. They are positively charged because of the free amino groups of five of their six 2,4-diaminobutyric acid (Dab) components. They are dissected into three characteristic structural elements. They are acylated at their N-terminal Dab residue either by S-6-methyl octameric acid or (S)-6-methyl heptanoic acid. Dab (1) is part of a linear tripeptide segment Dab(1)-Thr(2)- Dab(3)-. Dab (4) is involved in formation of a heptapeptide ring by condensation of its free amino group with the carboxyl group of the C-terminal Thr (10). The heptapeptide ring harbors a variable motif at positions R(6) and R(7). These residues appearing in D-configuration are specific for the polymyxin type and can therefore be used for the classification of these antibiotics. The therapeutically most important species are polymyxin B [17] and polymyxin E (colistin) [33,34,35]. Polymyxins E are produced by *P. polymyxa* DSM 32871. Here, positions 6 and 7 are occupied by two leucine residues, while for polymyxin B Leu (6) is replaced by Phe. The corresponding motif of polymyxin P produced by strain M1 is D-Phe-D-Thr [36,37]. Polymyxin B and E species showing numerous variations both in their fatty acid side chain as well as by amino acid substitutions in their peptide portion were extensively characterized by liquid chromatography electrospray ionization ion trap tandem mass spectrometry [37,38,39,40].

Biosynthesis of fusaricidins and polymyxins is accomplished nonribosomally at multifunctional protein templates. Fusaricidin formation is encoded by the *fus* gene cluster. Fusaricidin synthetase FusA, the gene product of the central *fusA* gene, consists of six modules for activation and condensation of all six amino acid components forming the complete peptide chain of fusaricidin [10].

Gene clusters *pmx* responsible for the biosynthesis of polymyxins A, B, E and P were identified and sequenced from the genomes of their producer organisms [41,42,43,44]. In each case *pmx* comprises genes for three peptide synthetases and two putative ABC transporters. Polymyxin synthetase is a complex of three multienzymes PmxA, B and E. PmxE is a five module multienzyme, activating and condensing the first five amino acid components of polymyxins forming the pentapeptide intermediate L-Dab-L-Thr-D-Dab-L-Dab-L-Dab-. PmxA contains four modules specific for the next four amino acid substrates extending the growing peptide chain until the nonapeptide. Finally, polymyxin synthesis is terminated by PmxB containing the module specific for the C-terminal L-Thr residue together with a thioester domain. The structural complexity of lipopeptides, such as fusaricidins and polymyxins, is not obvious from the gene level, but depends on the relatively high substrate variability at the reaction centers of the involved NRPS producer multienzymes.

Polymyxins are in focus as last resort antibiotics against multiresistant Gram-negative pathogens, such as the majority of *Enterobacteriaceae, K. pneumoniae, A. baumannii* or *P. aeruginosa* [45,46]. The therapeutic application of polymyxins certainly is the most important public interest, but beyond this there are also promising aspects for their usage as biocontrol agents in plant biotechnology. Recently, polymyxin P was demonstrated to suppress phytopathogenic *Erwinia* spp., the causative pathogen of fire blight and soft rot diseases [44].

Another class of attractive microbial secondary metabolites exhibiting manifold important activities are volatile organic compounds (VOCs), which display high chemical diversity [47,48,49]. They gain increasing attention particularly concerning plant growth promotion [50,51,52], biocontrol against bacterial and fungal pathogens [51,52] and induction of systemic resistance against phytopathogens [53,54,55].

In our work we performed comprehensive studies on the variety of the fusaricidin and polymyxin lipopeptide complexes produced by *P. polymyxa* DSM 32871 and M1 by MALDI-TOF mass spectrometry. These antifungal and antibacterial agents are of high relevance for inactivation of both human and plant pathogens. In addition, the diversity of volatile organic compounds (VOCs) released by these strains was studied with a specific headspace GC–MS analysis method under different growth conditions.

## 2. Results

In this work we investigated the biosynthetic potential of the *Paenibacillus polymyxa* strain DSM 32871. Its product pattern was compared with that of the well characterized M1 strain [10,11], which was implemented as a model strain. To explore the antimicrobial capacity of these strains we investigated their biosynthetic products by MALDI-TOF mass spectrometry in a time and space dependent fashion. We focused on two lipopeptide complexes, the fusaricidins and the polymyxins, as well as their volatiles which facilitate long-range interactions with plants and their native microbiota in the rhizosphere.

For this purpose, DSM 32871 and M1 were grown in various cultivation media (Landy, LOP, LB, TSA and GSC). Samples were taken at growth times of 12; 24; 48; 72 and 96 h to investigate the time profile of the production of the tested bioactive compounds and to determine their specific location. It was assessed whether they were residing in the cell or at the outer cell wall, or if they were released into the cultivation media. 

Figure 1 shows MALDI-TOF mass spectra of surface extracts (Figure 1A,C) and culture filtrates (Figure 1B,D) of DSM 32871 and M1 in the mass range of 800–2000 Da indicating three lipopeptide complexes, the fusaricidins, polymyxins and tridecaptins. The strains were cultivated in the LOP medium. Surface extracts were prepared by extraction of cell material obtained from agar plates with 50% ACN/0.1% TFA. Fusaricidins and tridecaptins preferentially were found attached to the outer surface of the *Paenibacillus* strains, while the polymyxins were mainly released into the cultivation media. At growth times higher than 48 h fusaricidins were also enriched in the culture filtrates.

In particular, the fusaricidin complex is distinguished by an unusually high variability [11]. Its composition depends on the growth medium as is apparent from Appendix A (Appendix A) for cultivation of DSM 32871 and M1 in the Landy/LOP and LB medium. Interestingly, for DSM 32871 grown in the Landy or LOP medium in the MALDI-TOF mass spectra (Appendix A and Figure 2A), fusaricidins A and B with molecular masses of *m*/*z* = 883.9 and 897.9 and their derivatives at *m*/*z* = 954.9 and 968.9 differing from the parent ions by a mass difference of 71 Da were prevalent, while for DSM 32871 grown in the LB medium and for M1 cultivated in both the Landy/LOP and the LB media fusaricidins A, B, C (*m*/*z* = 947.9) and D (*m*/*z* = 961.9) were the prominent species.

The culture filtrate of DSM 32871 grown for 48 h in the LOP medium (Figure 2A) was fractionated by reversed phase HPLC on a 300 SB-C8 Zorbax column. The chromatogram in Figure 2B shows product fractionation in fractions 20 to 30. MALDI-TOF mass spectra of aliquots of fractions 22–30 is shown in Appendix A (Appendix A) demonstrating the separation of the produced compounds. Fusaricidins are the main products of DSM 32871. They were distributed among a broad range in fractions 22–30. Polymyxins appeared in fractions 22 and 23. Tridecaptins were found in fractions 28–30.

The fusaricidins produced by *P. polymyxa* DSM 32871 are summarized in Table 1. They comprise the well-known fusaricidins A-F [5,6,10,11]. In addition, fusaricidins were found at *m*/*z* = 869.53 and 855.51 which could be derived from fusaricidin A either by substitution of Thr (1) or both threonines in positions 1 and 4 by serines, respectively. Apparently, three series of new fusaricidin species were detected. For DSM 32871 the occurrence of derivatives of all these fusaricidins was characteristic, which differ from their parent ions by a mass difference of 71 Da indicating modification by attachment of an alanine residue. These variants were among the dominating species in the mass spectra of DSM 32871. In addition, for this strain two other yet unknown fusaricidins with molecular masses of *m*/*z* = 899.6 and 913.6 were found. They are related to fusaricidins A and B with a mass difference of 16 Da, indicating a hydroxylation of these species. In addition, for these new fusaricidins +71-derivatives (*m*/*z* = 970.5 and 984.5) appeared. All these variants were either not found or are present in only low amount in samples of the reference strain M1. The structure of all these new fusaricidins were investigated by fragment analysis using LIFT-MALDI-TOF/TOF mass spectrometry [56]. Fragment spectra for fusaricidin A (*m*/*z* = 883.5) (A) and related compounds (*m*/*z* = 954.6 (B) and 899.6 (C)) are shown in Figure 3A–C.

In previous studies [10,11] we demonstrated that by laser excitation fusaricidins preferentially are decomposed in ß-position to the carboxyl group of their fatty acid residue forming two characteristic *fragments a and b* according to the fragment pattern displayed in Figure 1 for ring-opened fusaricidin A:

*Fragment a* functions as an indicator for the fatty acid constituent of fusaricidins which is shortened by the two carbon atoms at its C-terminal end, which are connected via the carboxyl group of the fatty acid component with the amino group of the starter amino acid being part of *fragment b* containing the complete peptide chain. In Figure 3 the mass peak of *fragment a* was strongly dominating in the product ion spectra of fusaricidins, while *fragment b* appeared with substantially lower intensity. *Fragment b* was visualized in the expanded mass region from *m*/*z* = 600–720. The molecular masses for the fusaricidins produced by *P. polymyxa* DSM 32871 together with the mass data for the corresponding *fragment ions a and b* derived from the product ion spectra are summarized in Table 1.

In Figure 3A,B the fragment spectrum of fusaricidin A with a C15-guanidino-pentadeca-3-hydroxy fatty acid chain is shown. As is apparent from Figure 1 [11] here the mass numbers of *fragments a and b* amount to *m*/*z* = 256.3 and 628.3, respectively. For the +71-derivative the same value for *fragment a* was found as for the unsubstituted fusaricidin A (Figure 3C), but *fragment b* as apparent in Figure 3D was 71 mass units higher (*m*/*z* = 699.2) indicating that the modification is located in the peptide portion. In contrast, for the +16-variant of fusaricidin A (*m*/*z* = 899.6) *fragment a m/z* = 272.2 (Figure 3E) was 16 mass units higher than for the unsubstituted fusaricidin A, while the peptide part *(m/z* = 628.3 for *fragment b*) remained unchanged (Figure 3F). Apparently, here the fatty acid part of the derivative is hydroxylated.

By evaluation of the LIFT-TOF/TOF product ion spectra of the fusaricidins produced by DSM 32871 listed in Table 1, a third group of yet unknown variants was found with molecular masses *m*/*z* = 897.3 and 911.2, which revealed as derivatives of fusaricidins A and B, exhibiting a value of *m*/*z* = 270.3 for *fragment a*. According to the data for *fragment b* of *m*/*z* = 628.3 and 642.2 they can be attributed to fusaricidins A and B bearing a C16-guanidino-3-hydroxy-hexadecameric acid as fatty acid constituent.

In Figure 4A–C the complete sequence of fusaricidin A and its +71- and +16-variants obtained from series of b_n_- and y_n_-fragment ions are demonstrated. Indeed, unsubstituted fusaricidin A (Figure 3A) and its +16-derivative (Figure 3C) show the same peptide sequence, while the +71-variant (Figure 3B) is substituted at its allo-threonine residue in position 4 by an alanine forming an ester bond with its free hydroxyl group. This feature is corroborated by the sequence of the corresponding *fragments b* of fusaricidin A (Figure 5A) and its +71-derivative (Figure 5B). These results were substantiated by the product ion spectra in Appendix A (Appendix A) showing their upper part in expanded version. Here, the value of fragment b_4_ (*m*/*z* = 769.14) is 71 mass units higher than b_4_ of the unmodified fusaricidin A, indicating substitution of allo-Thr in position 4 by an alanine. Correspondingly, b_5_ (*m*/*z* = 812.63) is shifted to *m*/*z* 883.66 in the case of the +71-variant with a molecular mass of *m*/*z* = 954.6. This result is compatible with previous data obtained for fusaricidins C and D [10].

In Figure 6 and Table 2 the polymyxin products of the two investigated *P. polymyxa* strains are shown. DSM 32871 produces two isomers of polymyxin E (E1 and E2) with molecular masses [M + H]^+^ = 1155.7 and 1169.8 (see Table 2), while M1 forms two variants of polymyxin P (P1 and P2) with masses of [M + H]^+^ = 1177.4 and 1191.4 (Figure 6C and Table 2) [38,44]. For both strains the isomers differ by a C8 or a C9 fatty acid. The protonated forms of these polymyxin families together with their alkali adducts are summarized in Figure 6 and Table 2. Both polymyxins have a high affinity for potassium, as shown by the [M − H + 2K]^+^-ions indicating binding of two potassium atoms.

The structure of polymyxins E studied by LIFT-MALDI-TOF/TOF MS is presented in Figure 7. Product ion spectra were taken either starting from the fatty acid constituent or the first amino acid Dab (1) (Figure 7A) as well as from Dab (5) and Leu (6) (Figure 7B) modelling the polymyxin structure from series of b_n_- and y_n_-ions. The fatty acid residues show masses of *m*/*z* = 126.1 and 140.1, respectively. The obtained sequence of polymyxin E is in accordance with the data reported in the literature [38,39,40,41,42]. The structure of polymyxin P was presented in a previous paper [36,44].

Interestingly, both for polymyxin E and P we observed a family of minor species apparent from Figure 6 at the higher mass range. For example, mass peaks were found at *m*/*z* = 1302.0 and 1316.0 for polymyxin E as well as at *m*/*z* = 1323.9 and 1337.9 for polymyxin P each showing a mass difference of approximately 146 Da relating to the protonated unsubstituted forms of E1/E2 and P1/P2, respectively (see Table 2). These species were sequenced by LIFT-MALDI-TOF/TOF MS. All of them showed the polymyxin sequence with a substitution of the Dab residue in position 4 by a threonine in combination with yet unknown structural elements at the C-terminal end of approx. 146 or 197 Da. This replacement was corroborated by nearest neighbor studies in Table 3 modelling the polymyxin structure by di-, tri-, tetra- and pentapeptide analysis.

For example, in Figure 8 sequencing of the species with a molecular mass of *m*/*z* = 1316.0 is demonstrated which is a derivative of polymyxin E2 modified at its C-terminal end with a residue of approximately 146 Da. Similar results were achieved for the corresponding E1 derivative with a molecular mass of *m*/*z* = 1302.0. These compounds show a higher hydrophobicity than the original polymyxins demonstrated in Figure 6 and Figure 7. In the HPLC-profile (Figure 2B) they were found in fractions 28 and 29 together with the tridecaptins, while the unsubstituted polymyxins E appeared in fraction 23. The nature of the additional elements in the sequence of the minor species have still to be elucidated.

Plant growth promoting rhizobacteria and endophytes, like *Bacilli* and *Paenibacilli*, produce a wide spectrum of volatile organic compounds (VOCs), which exhibit a high chemical diversity and manifold activities. In this work we investigated the volatilomes of *Paenibacillus polymyxa* DSM 32871 and M1, which were grown in four different cultivation media (NA, LB, TSA and Landy). The obtained results summarized in Table 4 show that the production of VOCs by such organisms is strain and medium dependent.

The identification of microbial volatile organic compounds (mVOCs) was performed via HS-SPME GC–MS, compared to the NIST MS 14 database and confirmed via Kovats Index. In total, 25 different volatile compounds could be identified in the headspace of *P. polymyxa* M1. In contrast to M1, only 20 volatile substances were detected as products of *P. polymyxa* DSM 32871. Interestingly, M1 exhibited the broadest array of mVOCs when grown on NA, whereas DSM 32871 produced the highest number of volatiles cultivated in TSA. Both strains produced a mixture of ketones, alcohols, alkanes, alkenes, and a variety of alkylpyrazines. The volatilomes of *P. polymyxa* DSM 32871 and M1 are highly overlapping with n-hexane being the sole compound, which could only be found in DSM 32871. Five compounds were unique to M1: 2-methyl-1-propanol, 2-methylbutanenitrile, dimethyl sulfone, butyl acetate, 3-methylbutyl acetate and 2-heptanone. In comparison to other plant growth promoting rhizobacteria of the order Bacillales, such as *Bacillus subtilis*, *Bacillus atrophaeus*, *Bacillus amyloliquefaciens*, *Bacillus velezensis*, and *Bacillus licheniformis* [57], both *P. polymyxa* strains emitted a variety of alkylpyrazines. In previous studies, alkylpyrazines exhibited antibacterial, antifungal, antimycobacterial, algicidal and antiviral activities [58,59,60,61]. Additionally, 2-methyl-1-butanol, 2-heptanone and 2-ethyl-1-hexanol revealed growth inhibition against *Agrobacterium tumefaciens* C58 and *Synechococcus *sp. PCC 7942 as well as fungal pathogens, such as *Phyllosticta citricarpa* and *Aspergillus fumigatus* [62,63,64]. Acetoin, 2,3-butanediol and 3-methyl-1-butanol can be found in the volatilomes of DSM 32871 and M1. These compounds are known for their plant growth promoting effect and induce systemic resistance (ISR) in distinct plants [50,54,65,66,67,68]. Examples for medium specificity included 2,3-butanediol which was formed by both strains only in the Landy medium, while 1-octen-3-ol and 2-ethyl-1-hexanol have been detected specifically in the TSA-medium. *Paenibacillus polymyxa* is a potent producer of 2,3-butanediol which is often cogenerated by other volatile products, such as ethanol, acetoin and acetic acid together with specific exopolysaccharides [69]. In particular, 2,3-butanediol is of high commercial importance for wide industrial applications as next generation fuels and as a feedstock compound for pharmaceuticals, cosmetics and foot preservatives.

The so far known biological functions of the observed volatiles are summarized in the right part of Table 4 [50,54,62,64,65,66,67,70,71,72,73,74]. However, their biological effects and their underlying mode of action still need to be investigated.

## 3. Discussion

In this work we investigated the biosynthetic potential of the *P. polymyxa* strains DSM 32871 and M1. Both strains produce among some other bioactive compounds three lipopeptide complexes, the fusaricidins, the polymyxins and the tridecaptins. Our efforts were focused on the mass spectrometric analysis of the fusaricidins and polymyxins formed by the DSM 32871 strain, which were compared with those produced by the well characterized M1 strain [10,11,44]. Both lipopeptide families are of high commercial interest because of their potential importance for applications both in agrobiotechnology and medicine.

The fusaricidins being the main products of DSM 32871 and M1 are known to exert potent antifungal activities. They show an unusual complexity because of manifold variations both in their peptide moiety as well as in their fatty acid part. For the reference strain M1 more than 80 variants have been reported so far [11]. Interestingly, by MALDI-TOF mass spectrometric analysis of the fusaricidin complex of strain DSM 32871 three novel series of yet unknown fusaricidins were detected specifically when cultivated in the Landy/LOP medium. Prominent mass peaks of high intensity were found for fusaricidin variants modified at the free hydroxyl group of their allo-Thr component in position 4 by esterification with an alanine residue. Small amounts of such species have previously been found also for strain M1 specifically for fusaricidin C and D [10]. Another set of novel fusaricidins was detected, which are modified by an additional OH-group in their fatty acid part, which still remains to be precisely located.

Fusaricidins have been utilized for biocontrol of numerous fungal plant diseases [26,27,29,30,31]. The aim of future studies will be the purification and provision in semi/preparative scale of selected fusaricidins by high resolution separation techniques for biocontrol applications in plant biotechnology.

Another prominent lipopeptide family produced by *P. polymyxa* DSM 32871 and M1 are the polymyxins which are cyclic decapeptides. They are well suited to inactivate Gram-negative pathogens, such as *Klebsiella*, *Enterobacteriaceae*, *A. baumannii* and *P. aeruginosa* [44,45,46]. DSM 32871 produces polymyxins E (colistins), while M1 forms polymyxins P. Polymyxins were discovered in 1947 [76] and used for at least two decades as efficient antibiotics. Due to nephrotoxic and neurotoxic side effects they were taken from the market; however, recently polymyxins were retrieved again as last-resort antibiotics to counteract multiresistant super bugs [32,37,45,46]. High financial investment is made in several countries to optimize their structure, activities and pharmacokinetic properties with the aim to create novel efficient agents to contribute to the pertinent problem to overcome antibiotic multiresistance, which is one of the most urgent issues to protect mankind against otherwise invincible dangerous pathogens.

By MALDI-TOF mass spectrometric analysis of the polymyxin E and P complexes produced by DSM 32871 and M1 we detected modified minor polymyxin species (see Table 2), which would be worth testing therapeutically. For example, for both polymyxin species we found variants which showed the polymyxin E or P sequence, but in addition contain a residue of 146 Da which could not yet be identified. We hypothesize that they may be glycosylated. The aim of our forthcoming studies will be to isolate and upgrade these variants in pure form by high resolution separation methodology for enabling structure elucidation and clinical testing.

Another important aspect of polymyxins is their application as biocontrol agents in agrobiotechnology to inactivate deleterious phytopathogens in order to prevent and cure plant diseases caused by Gram-negative plant pathogenic bacteria. For example, polymyxins P produced by the M1 strain were recently applied successfully against *Erwinia* strains [44].

In the third part of our work, we investigated the manifold volatile organic compounds (VOCs) produced by *P. polymyxa* strains DSM 32871 and M1. VOCs attain increasing interest concerning their influence on the plant microbiomes and the communication between their members in the rhizosphere as well as their action inside plant tissues. Studies have been initiated to use VOCs produced by *P. polymyxa* strains as biocontrol agents to inactivate phytopathogenic microorganisms and to induce systemic resistance against such pathogens [77,78,79]. The wide spectrum of volatiles produced by DSM 32871 and M1 is presented in Table 4. They comprise ketones, alcohols, alkanes and alkenes. A characteristic feature of different *Paenibacillus* species is their production of volatile pyrazine derivatives [58,59,60,61].

Among the volatiles produced by *Paenibacillus polymyxa* 2,3-butanediol is of particular commercial importance for wide industrial applications, such as next generation fuels and as a feedstock compound for pharmaceuticals, cosmetics and food preservatives. Downstream processing in 2,3-butane fermentation is strongly hampered by the concomitant production of exopolysaccharides which increase medium viscosity. This problem can be avoided by disruption of the gene encoding levansucrase, the major enzyme responsible for EPS biosynthesis creating a levansucrase null mutant [69]. On the other hand, extracellular polysaccharides play an important role in biofilm formation of *Paenibacilli* for successful colonization of plant roots in the rhizosphere. EPS are responsible for the contact between bacteria and root surfaces for effective development of plant growth promotion and biocontrol [80,81].

The specific equipment of the investigated *P. polymyxa* strains with VOCs revealed to be strain and medium dependent. Here, the prospect of forthcoming studies is the investigation of their mechanism of action within the plant microbiome.

As a resumé of our detailed analyses of *P. polymyxa* strains we emphasize prospecting their potential for applications both for medical uses as well as in plant biotechnology.

## 4. Materials and Methods

### 4.1. Materials

The matrices α-cyanohydroxycinnamic acid (CCA) and dihydroxy-benzoic acid (DHB) used for MALDI-TOF MS were obtained from Bruker (Bremen, Germany). Acetonitrile (ACN, HPLC grade) was purchased from Merck (Darmstadt, Germany), trifluoroacetic acid (TFA) from Sigma-Aldrich (Deisenhofen, Germany).

### 4.2. Cultivation of Organisms

For the preparation of surface extracts of *Paenibacillus polymyxa* strains DSM 32871 and M1 were grown on agar plates using three cultivation media [57]: (a) Lysogeny broth (Luria–Bertani) (LB) medium, (b) Landy medium [82] and (c) TSA medium solidified with 1.5% agar in Petri dishes for 24, 48 and 72 h at 30 °C. In addition, liquid fermentations were carried out in 100 mL Erlenmeyer flasks at 30 °C and 200 rpm in an orbital shaker (Buehler, Germany) in the LOP and LB medium to detect products released by the strains into the culture medium. The LOP medium is a modified Landy medium which compared with the original version [82] contains higher amounts of glucose × H_2_O (42 g/L) and Na-glutamate × H_2_O (14 g/L).

### 4.3. Sample Preparation

In order to obtain complete profiles of bioactive peptides produced by the investigated *P. polymyxa strains* DSM 32871 and M1, their products were detected by MALDI-TOF MS (a) in surface extracts of cells picked either from agar plates or harvested from liquid cultures by centrifugation for 10–20 min at 15,000 rpm, (b) in culture supernatants after growth for 12; 24; 48; 72 and 96 h and (c) after cells disintegration by solubilization with 80% trifluoroacetic acid. For (a), a wire loop of cell material was picked, suspended in 50 µL 50% acetonitrile/0.1% TFA and extracted for 15 min by vigorous vortexing. Finally, cells were spun down at 15,000 rpm for 10 min.

### 4.4. HPLC Fractionation of the Bioactive Compounds

An amount of 330 µL of the culture filtrate of *P. polymyxa* DSM 32871 cultivated in the LOP medium was diluted with 660 µL 0.1% TFA. This sample was applied to a 300 SB-C8 Zorbax column (4.6 × 250 mm; 5 µm) and fractionated by reversed-phase HPLC using an Agilent (1200 series) instrument (Agilent Technologies, Waldbronn, Germany). Bioactive compounds were eluted by a two-step gradient from 0 to 70% eluent B in 70 min and from 70 to 95% eluent B in 5 min (70–75 min) followed by isocratic elution at 95% eluent B for 10 min at a flow rate of 0.5 mL/min. Eluent A was 0.1% TFA in water; eluent B was 99.9% ACN/0.1% TFA. Fractions of 1 mL were collected and evaporated to dryness in a SpeedVac evaporator (Uniequip, Martinsried, Germany). The dried material was dissolved in 30 µL 50% aqueous ACN/0.1 TFA and tested mass spectrometrically.

### 4.5. Profiling of Bioactive Peptides by MALDI-TOF MS

Bioactive peptides of *P. polymyxa* DSM 32871 and M1 were detected and identified by MALDI-TOF MS, as outlined previously [11,13,58]. A Bruker Autoflex Speed TOF/TOF mass spectrometer (Bruker Daltonik, Bremen, Germany) was used with Smartbeam laser technology using a 1 kHz frequency-tripled Nd-YAG laser (λ_ex_ = 355 nm). Samples (2 µL) of surface extracts and culture supernatants were mixed with 2 µL matrix solution (a saturated solution of α-hydroxy-cinnamic acid in 50% aqueous ACN containing 0.1% TFA), spotted on the target, air dried and measured. Mass spectra were obtained by positive-ion detection in reflector mode. Monoisotopic masses were obtained. Parent ions were detected with a resolution of 10,000. Sequence analysis of the lipopeptide products was performed by LIFT-MALDI-TOF/TOF mass spectrometry in laser induced decay (LID) mode [56]. The product ions in the LIFT-TOF/TOF fragment spectra were obtained with a resolution of 1000.

The mass spectrometry results did not allow optical isomers to be distinguished. Therefore, the configurations of the amino acid components were not indicated.

### 4.6. Identification of Microbial Volatiles Using GC–MS

*Paenibacillus polymyxa* strains DSM 32871 and M1 were grown in 20 mL headspace vials (75.5 × 22.5 mm; Chromtech, Idstein, Germany) filled with 8 mL of (a) nutrient agar (Sifin Diagnostic GmbH, Berlin, Germany), (b) Lysogeny broth (Luria–Bertani) (LB) medium, (c) Landy medium [82] and (d) TSA medium. Equal distribution was assured by parallel application of cell material in three vials per organism. After 24 h of incubation at 30 °C, the volatiles accumulating inside the vials were measured by headspace solid-phase microextraction gas chromatography–mass spectrometry (HS-SPME GC–MS). Compound separation and detection was performed on a system combining a gas chromatograph 7890A with a quadrupol mass spectrometer 5975C (Agilent Technologies, Waldbronn, Germany). A SPME fiber consisting of divinylbenzene/carboxen/polydimethylsiloxane (DVB/CAR/PDMS) was used for sampling (Sigma-Aldrich, St. Louis, MO, USA). HS samples were separated in a (5%-phenyl)methylpolysiloxane column (60 m × 0.25 mm i.d.; 0.25 µm film thickness; DB-5MS; Agilent Technologies, Waldbronn, Germany). Subsequently, electron ionization with 70 eV and detection in the mass range 25–350 Da were performed. The fiber was desorbed at 200 °C for 8 min to eliminate potential residues before initial measurements. The inlet temperature was set to 270 °C. The temperature gradient of the column was maintained at 70 °C for 1.5 min, raised to 200 °C at a rate of 16 °C/min and finally kept at 200 °C for 0.5 min. The helium flow rate was adjusted to 1.2 mL/min. For identification of the volatiles the received spectra were matched to the NIST Mass Spectral Database.

## Data Availability

The data presented in this article are available on request from the corresponding author.

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
