# Peer review of "Fusaricidins, Polymyxins and Volatiles Produced by Paenibacillus polymyxa Strains DSM 32871 and M1"

_pathogens, 2021, doi:10.3390/pathogens10111485_

Round 1

Reviewer 1 Report

This paper written by Pascal Mülner et al tested the production of fusaricidins, polymyxins and volatiles by Paenibacillus polymyxa strains by MALDI-TOF or GC mass spectrometry. Authors profiled bioactive peptides and identified the microbial volatiles.

This is very interesting research for future application in medical science or agricultural system.

In general, this paper was well written, and with interesting results.

It is great for the authors to prove that Paenibacillus polymyxa strains selected in this study can produce fusaricidins, polymyxins and volatiles, and their types.

I just have some minor comments:

For the title, it is no need to put the strain names. And it will be better to precise what is detected for fusaricidins, polymyxins or volatiles?

I was wondering what is the pathogen inhibition effect of Paenibacillus polymyxa strains used in this study? And what is the contribution of fusaricidins, polymyxins and volatiles produced by these strains to pathogen inhibition?

Authors wrote “The fusaricidins show potent antifungal activities and are distinguished by an unusual variability” in the abstract. Did authors sequence the whole genome of studied bacterial strains? Is there any link between the genomic information with this “unusual variability”?

Can authors detect how much (quantity) these products can be produced by selected Paenibacillus polymyxa strains growing in different media with different growth time?

Author Response

Response to the Reviewer 1 Comments

Response 1 – change of title

We changed the title as follows:

Fusaricidins, polymyxins and volatiles produced by Paenibacillus polymyxa strains as potent  biocontrol agents – a mass spectrometric study

Response 2:

As outlined in the Introduction fusaricidins provide a large reservoir of potent antifungal agents against phytopathogenic fungi, while polymyxins efficiently inactivate Gram-negative bacterial pathogens. Volatiles display manifold important activities, such as biocontrol agents against both bacterial as well as fungal pathogens. All these bioactive compounds show promising aspects for pathogen inactivation both in medicine and plant biotechnology. The aim of our manuscript was to explore novel variants of these agents by high resolution mass spectrometric techniques and to provide them for biocontrol experiments, which will the subject of future research as indicated in the Discussion.

Response 3:

The high variability of both fusaricidins and polymyxins is not obvious from the gene level. It is a matter of the relatively high substrate variability at the reaction centers of the involved NRPS multienzymes.

The reference strain P. polymyxa M1 has well been characterized as an efficient plant growth  promoting rhizobacterium. Here the complete genome was sequenced by Niu et al. In contrast, DSM 32871 is a novel strain. Here the genome sequence is not yet available.  We supplement this information in the Introduction.

Response 4:

At present we could not quantify our products. For this purpose specific methodology has to be developed. This will be one of the tasks for our future work. Our mass spectrometric measurements indicate that for both strains fusaricidins are the main products in any of the used growth media appearing in similar amounts for strains DSM 32871 and M1. Fusaricidins were preferentially  attached to the outer surface of Paenibacillus strains, while the polymyxins were mainly released into the culture media, in particular, at growth times lower than 48 h. The medium dependence of the volatiles is outlined in the Results.

Reviewer 2 Report

  1. For 2,3-butanediol listed in Table, why referencing Han et al., 2006 that described GacS-dependent production of 2R, 3R butanediol by Pseudomonas chlororaphis 06? A reference on 2,3-butanediol production by Paenibacillus polymyxa should have been provided instead.
  2. Paenibacillus polymyxa is a good producer of exopolysaccharides. Why was it not mentioned in the manuscript? Other compounds produced by Paenibacillus polymyxa include ethanol, acetoin, acetic acid, and 2,3-butanediol. At least a sentence should be included in the introduction section highlighting these compounds to give readers a broad view of biotechnological significance of Paenibacillus polymyxa. A good reference that covered these compounds including exopolysaccharides is ‘Okonkwo et al. (2020) Inactivation of the Levansucrase Gene in Paenibacillus polymyxa DSM 365 Diminishes Exopolysaccharide Biosynthesis during 2,3-Butanediol Fermentation. Applied and Environmental Microbiology 86 (9): e00196-20. doi: 10.1128/AEM.00196-20’
  3. Although the research work focused on the mass spectrometric analysis of lipopeptide complexes produced by Paenibacillus polymyxa, the different amounts/quantities produced by the two strains, DSM 32871 and M1, should be presented.
  4. Using mass spectrometry, some unknown variants of the lipopeptide complexes were identified. What is the implication of these variants in relation to the antifungal and antibiotic properties of Fusaricidins and polymyxins?
  5. There are some minor grammatical errors that need to be corrected throughout the manuscript. For examples: A) line 21, ‘For both strains novel’ should be replaced with ‘Both strains are novel’; B) line 23, ‘attain increasing’ should be replaced with ‘have gained’; C) line 116, replace ‘disintegration cells’ with ‘cells disintegration’
  6. In some places, P. polymyxa was not written in italics and semicolon was used instead of commas. Please check the manuscript and make corrections.
  7. In line 140, you wrote 10.000; do you mean 10,000?

Author Response

Response to the Reviewer 2 Comments

Response 1:

A reference on 2,3-butanediol production specific for Paenibacillus polymyxa is provided, but we retain the reference Han et al., 2006 on the right side of Table 4. Here the general knowledge of the so far known biological functions of the detected volatiles are summarized.

Response 2:

The production of exopolysaccharides by Paenibacillus polymyxa is highlighted under Results and in the Discussion. The proposed references are integrated into the text.

Responses 3 and 4:

The central aim of the research presented in our manuscript was to explore novel biocontrol agents by high resolution mass spectrometric techniques for future applications in pathogen inactivation both in medicine and plant biotechnology. At present we cannot quantify the detected products of strains DSM 32871 and M1. For this purpose specific efficient methodology has to be developed.  In particular, the products have to be provided in pure form which is a difficult task because of the high complexity of the product mixtures. This requires application of the full spectrum of separation methodology. For these reasons there are no inactivation and biocontrol experiments with our products available yet. Quantification and microbial testing of the products will be tasks for our future research, as outlined in the Discussion.

Response 5:

Grammatical errors have been corrected in the manuscript

Response 6:

The corrections were made. Commas were used throughout the manuscript instead of semicolons.

Response 7:

10.000 is correct!

Reviewer 3 Report

Greetings dear colleagues, 

In this work you have characterized nonribosomal peptides and volatiles compounds produce by two Paenibacillus micro-organisms (DSM32871 and M1). You tested several growth conditions and purifications steps (membrane-linked/supernatant) in order to improve the recovery of the targeted molecules.
The non ribosomal peptides studied belong to the fusaricidins and polymyxins. The former have anti Gram+ and fungi activities, whereas the latter are anti-Gram -. Two polymyxins (B and E= colistins) are used for antibiotic treatments. 

Things to improve :

1) The work looks like a lot to an update of the Govaerts paper of 2002 (quote un the references) using last mass spectrometric equipments. Does the new data acquired in 2021 add knowledge regarding the "old" Govaerts work ?
By the way some references are lacking regarding polymyxins variants :
- Shaheen M, Li J, Ross AC, Vederas JC, Jensen SE. Paenibacillus polymyxa PKB1 produces variants of polymyxin B-type antibiotics. Chem Biol. 2011 Dec 23;18(12):1640-8. doi: 10.1016/j.chembiol.2011.09.017. PMID: 22195566.
-Govaerts C, Orwa J, Van Schepdael A, Roets E, Hoogmartens J (2002a) Characterization of polypeptide antibiotics of the polymyxin series by liquid chromatography electrospray ionization ion trap tandem mass spectrometry. J Pept Sci 8:45–55
-Govaerts C, Orwa J, Van Schepdael A, Roets E, Hoogmartens J (2002b) Liquid chromatography-ion trap tandem mass spectrometry for the characterization of polypeptide antibiotics of the colistin series in commercial samples. J Chromatogr A 976:65–78
-Orwa JA, Govaerts C, Gevers K, Roets E, Van Schepdael A, Hoogmartens J (2002) Study of the stability of polymyxins B1, E1 and E2 in aqueous solution using liquid chromatography and mass spectrometry. J Pharm Biomed Anal 29:203–212

2) The paper is relatively long albeit being easy to read. 

3) In the introduction/discussion, the authors totally forgot to mention the numerous paper unravelling the biosynthetic pathways involved in the production of the non ribosomal peptides e.g.:

  • Tambadou F, Caradec T, Gagez AL, Bonnet A, Sopéna V, Bridiau N, Thiéry V, Didelot S, Barthélémy C, Chevrot R. Characterization of the colistin (polymyxin E1 and E2) biosynthetic gene cluster. Arch Microbiol. 2015 May;197(4):521-32. doi: 10.1007/s00203-015-1084-5. Epub 2015 Jan 22. PMID: 25609230.

  • -Choi SK, Park SY, Kim R, et al. Identification of a polymyxin synthetase gene cluster of Paenibacillus polymyxa and heterologous expression of the gene in Bacillus subtilis. J Bacteriol. 2009;191(10):3350-3358. doi:10.1128/JB.01728-08

It seems weird to be focussed on biochemical products structures, ignoring totally their way of synthesis. Indeed, the nonribosomal synthetase are especially involved in the chemical diversity of the non ribosomal peptides released. The lability of some of their A domains allows this variations. The authors should add a paragraph dealing with this aspects within at least the introduction in order to highlight their own work.

4) Relationships between culture medium/growth conditions and the production of nonribosomal peptides and volatiles compounds should be exploited a bit more. 

Why this scientific paper is interesting :

1) this work shows biochemical structures of top interest regarding the emergence of microbial pathogens that resist to the classical antibiotic treatments. 
2) Numerous polymyxins derivatives obtained by chemical synthesis exists. Those entities were developed in order to reduce the secondary effects of polymyxins B and E, and to enhance their antimicrobial properties. Nevertheless, one can not say the same regarding fusaricidins. These work enlarge the spectrum of the known variants of these latter. This may inspired the synthesis of some new chemical derivatives. 
3) I was not aware of the volatiles produced by Paenibacilli and their role in the control of the growth of some fungi. 
4) Specialists of mass spectrometry experiments should find useful data regarding the fragmentation pattern of non ribosomal peptides
5) The paper is easy to read and clear despite the important amount of data presented.

Misprints:
L13 Paenibacillus : the "P" is not in italic ?
L15 "inactivation potential": could you rephrase it ? 
L25 (25) space and
L83 the hydrophobic motif of polymyxin P ? what do you mean ? 
L186 as can be seen: could you rephrase it ? 
L171 L189 why fragment is in italic ? 
L188 why using resp instead of respectively ? 
L214 same remark for "Figs"
Discussion
L38 nown => known
L40 for the(space)reference 
L58 P.(space)Polymyxins
L58 "Stansly et al" ? 
L61 counteract
L105 .(space)In addition

good luck for the corrections

Author Response

Response to the Reviewer 3 Comments

Response 1 and 3:

Our work is not an update of the Govaerts paper! Govaerts et al. have made an extensive study of polymyxin B- and E-type antibiotics by a combination of liquid chromatography and high resolution mass spectrometry. In this way these authors detected numerous variants of these polymyxin species with variations both in their fatty acid side chain as well as in their peptide portion. In our work we identified so far completely unknown variants of polymyxins E and P with a specific substitution. They appear in a higher mass range than the unmodified species between m/z = 1300 – 1370. We hypothesize that they are glycosylated. There is no report of such species in the literature so far.

By the way, a major part of our work is dedicated to the fusaricidins which show an unusual complexity. But many thanks for the Govearts papers which we will address in the Introduction together with the pathways involved in the biosynthesis of fusaricidins and polymyxins. Indeed, the complexity of the biosynthetic products depends on the relatively high substrate variability at the reaction centers, such as the A-domains of the NRPS multienzymes, for example. The suggested references we will integrate into the text.

Response 2

There are so many results that we need the space for presentation.

Response 4

Our mass spectrometric measurements indicate that for both strains fusaricidins are the main products in any of the used growth media appearing in similar amounts for strains DSM 32871 and M1. Fusaricidins were preferentially attached to the outer surface of Paenibacillus strains, while the polymyxins were mainly released into the culture media, in particular at growth times lower than 48 h. The medium dependence of the volatiles is outlined in the Results.

Misprints have been corrected.

Throughout the text the characteristic fragments a and b of fusaricidins were indicated in italic.